# Persistence Fisher Kernel: A Riemannian Manifold Kernel for Persistence Diagrams

**Tam Le**
RIKEN Center for Advanced Intelligence Project, Japan
`tam.le@riken.jp`

**Makoto Yamada**
Kyoto University, Japan
RIKEN Center for Advanced Intelligence Project, Japan
`makoto.yamada@riken.jp`

## Abstract

Algebraic topology methods have recently played an important role for statistical analysis with complicated geometric structured data such as shapes, linked twist maps, and material data. Among them, *persistent homology* is a well-known tool to extract robust topological features, and outputs as *persistence diagrams* (PDs). However, PDs are point multi-sets which can not be used in machine learning algorithms for vector data. To deal with it, an emerged approach is to use kernel methods, and an appropriate geometry for PDs is an important factor to measure the similarity of PDs. A popular geometry for PDs is the *Wasserstein metric*. However, Wasserstein distance is not *negative definite*. Thus, it is limited to build positive definite kernels upon the Wasserstein distance *without approximation*. In this work, we rely upon the alternative *Fisher information geometry* to propose a positive definite kernel for PDs *without approximation*, namely the Persistence Fisher (PF) kernel. Then, we analyze eigensystem of the integral operator induced by the proposed kernel for kernel machines. Based on that, we derive generalization error bounds via covering numbers and Rademacher averages for kernel machines with the PF kernel. Additionally, we show some nice properties such as stability and infinite divisibility for the proposed kernel. Furthermore, we also propose a linear time complexity over the number of points in PDs for an approximation of our proposed kernel with a bounded error. Throughout experiments with many different tasks on various benchmark datasets, we illustrate that the PF kernel compares favorably with other baseline kernels for PDs.

## 1 Introduction

Using algebraic topology methods for statistical data analysis has been recently received a lot of attention from machine learning community [Chazal et al., 2015, Kwitt et al., 2015, Bubenik, 2015, Kusano et al., 2016, Chen and Quadrianto, 2016, Carriere et al., 2017, Hofer et al., 2017, Adams et al., 2017, Kusano et al., 2018]. Algebraic topology methods can produce a robust descriptor which can give useful insight when one deals with complicated geometric structured data such as shapes, linked twist maps, and material data. More specifically, algebraic topology methods are applied in various research fields such as biology [Kasson et al., 2007, Xia and Wei, 2014, Cang et al., 2015], brain science [Singh et al., 2008, Lee et al., 2011, Petri et al., 2014], and information science [De Silva et al., 2007, Carlsson et al., 2008], to name a few.

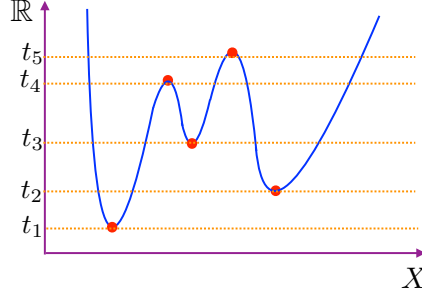

Figure 1: An illustration of a persistence diagram on a real-value function $f$. The orange horizontal lines are the boundaries of sublevel sets $f^{-1}((-\infty, t])$. For the 0-dimensional topological features (connected components), the topological events of births are happened at $t = t_1, t_2, t_3$ and their corresponding topological events of deaths are occurred at $t = +\infty, t_5, t_4$ respectively. Therefore, the persistent diagram of $f$ is $\mathrm{Dg}f = \{(t_1, +\infty), (t_2, t_5), (t_3, t_4)\}$.

In algebraic topology, *persistent homology* is an important method to extract robust topological information, it outputs point multisets, called *persistence diagrams* (PDs) [Edelsbrunner et al., 2000]. Since PDs can have different number of points, it is not straightforward to plug PDs into traditional statistical machine learning algorithms, which often assume a vector representation for data.

**Related work.**　There are two main approaches in topological data analysis: (i) explicit vector representation for PDs such as computing and sampling functions built from PDs (i.e. persistence lanscapes [Bubenik, 2015], tangent vectors from the mean of the square-root framework with principal geodesic analysis [Anirudh et al., 2016], or persistence images [Adams et al., 2017]), using points in PDs as roots of a complex polynomial for concatenated-coefficient vector representations [Di Fabio and Ferri, 2015], or using distance matrices of points in PDs for sorted-entry vector representations [Carriere et al., 2015], (ii) implicit representation via kernels such as the Persistence Scale Space (PSS) kernel, motivated by a heat diffusion problem with a Dirichlet boundary condition [Reininghaus et al., 2015], the Persistence Weighted Gaussian (PWG) kernel via kernel mean embedding [Kusano et al., 2016], or the Sliced Wasserstein (SW) kernel under Wasserstein geometry [Carriere et al., 2017]. In particular, geometry on PDs plays an important role. One of the most popular geometries for PDs is the Wasserstein metric [Villani, 2003, Peyre and Cuturi, 2017]. However, it is well-known that the Wasserstein distance is not *negative definite* [Reininghaus et al., 2015] (Appendix A). Consequently, we may not obtain positive definite kernels, built upon from the Wasserstein distance. Thus, it may be necessary to *approximate* the Wasserstein distance to achieve positive definiteness for kernels, relied on Wasserstein geometry. For example, [Carriere et al., 2017] used the SW distance—an *approximation* of Wasserstein distance—to construct the positive definite SW kernel.

**Contributions.**　In this work, we focus on the implicit representation via kernels for PDs approach, and follow Anirudh et al. [2016] to explore an alternative Riemannian geometry, namely the Fisher information metric [Amari and Nagaoka, 2007, Lee, 2006] for PDs. Our contribution is two-fold: (i) we propose a positive definite kernel, namely the Persistence Fisher (PF) kernel for PDs. The proposed kernel well preserves the geometry of the Riemannian manifold since it is directly built upon the Fisher information metric for PDs *without approximation*. (ii) We analyze the eigensystem of the integral operator induced by the PF kernel for kernel machines. Based on that, we derive generalization error bounds via covering numbers and Rademacher averages for kernel machines with the PF kernel. Additionally, we provide some nice properties such as a bound for the proposed kernel induced squared distance with respect to the geodesic distance which can be interpreted as stability in a similar sense as the work of [Kwitt et al., 2015, Reininghaus et al., 2015] with Wasserstein geometry, and infinite divisibility for the proposed kernel. Furthermore, we describe a linear time complexity over the number of points in PDs for an approximation of the PF kernel with a bounded error via Fast Gauss Transform [Greengard and Strain, 1991, Morariu et al., 2009].

## 2　Background

**Persistence diagrams.**　Persistence homology (PH) [Edelsbrunner and Harer, 2008] is a popular technique to extract robust topological features (i.e. connected components, rings, cavities) on real-value functions. Given $f : X \mapsto \mathbb{R}$, PH considers the family of *sublevel sets* of $f$ (i.e.

$f^{-1}((-\infty, t]), t \in \mathbb{R}$) and records all *topological events* (i.e. births and deaths of topological features) in $f^{-1}((-\infty, t])$ when $t$ goes from $-\infty$ to $+\infty$. PH outputs a 2-dimensional point multiset, called *persistence diagram* (PD), illustrated in Figure 1, where each 2-dimensional point represents a lifespan of a particular topological feature with its birth and death time as its coordinates.

**Wasserstein geometry.** Persistence diagram Dg can be considered as a discrete measure $\mu_{\mathrm{Dg}} = \sum_{u \in \mathrm{Dg}} \delta_u$ where $\delta_u$ is the Dirac unit mass on $u$. Therefore, the bottleneck metric (a.k.a. $\infty$-Wasserstein metric) is a popular choice to measure distances on the set of PDs with bounded cardinalities. Given two PDs $\mathrm{Dg}_i$ and $\mathrm{Dg}_j$, the bottleneck distance $\mathcal{W}_\infty$ [Cohen-Steiner et al., 2007, Carriere et al., 2017, Adams et al., 2017] is defined as

$$\mathcal{W}_\infty \left( \mathrm{Dg}_i, \mathrm{Dg}_j \right) = \inf_\gamma \sup_{x \in \mathrm{Dg}_i \cup \Delta} \| x - \gamma(x) \|_\infty \,,$$

where $\Delta := \{ (a, a) \mid a \in \mathbb{R} \}$ is the diagonal set, and $\gamma : \mathrm{Dg}_i \cup \Delta \to \mathrm{Dg}_j \cup \Delta$ is bijective.

**Fisher information geometry.** Given a bandwidth $\sigma > 0$, for a set $\Theta$, one can smooth and normalize $\mu_{\mathrm{Dg}}$ as follows,

$$\rho_{\mathrm{Dg}} := \left[ \frac{1}{Z} \sum_{u \in \mathrm{Dg}} \mathbb{N}(x; u, \sigma I) \right]_{x \in \Theta} , \tag{1}$$

where $Z = \int_\Theta \sum_{u \in \mathrm{Dg}} \mathbb{N}(x; u, \sigma I) \mathrm{d}x$, $\mathbb{N}$ is a Gaussian function and $I$ is an identity matrix. Therefore, each PD can be regarded as a point in a probability simplex $\mathbb{P} := \left\{ \rho \mid \int \rho(x) \mathrm{d}x = 1, \rho(x) \geq 0 \right\}^1$. In case, one chooses $\Theta$ as an entire Euclidean space, each PD turns into a probability distribution as in [Anirudh et al., 2016, Adams et al., 2017].

Fisher information metric (FIM)[2] is a well-known Riemannian geometry on the probability simplex $\mathbb{P}$, especially in information geometry [Amari and Nagaoka, 2007]. Given two points $\rho_i$ and $\rho_j$ in $\mathbb{P}$, the Fisher information metric is defined as

$$d_{\mathcal{P}}(\rho_i, \rho_j) = \arccos \left( \int \sqrt{\rho_i(x) \rho_j(x)} \mathrm{d}x \right). \tag{2}$$

## 3 Persistence Fisher Kernel (PF Kernel)

In this section, we propose the Persistence Fisher (PK) kernel for persistence diagrams (PDs).

For the bottleneck distance, two PDs $\mathrm{Dg}_i$ and $\mathrm{Dg}_j$ may be two discrete measures with different masses. So, the transportation plan $\gamma$ is bijective between $\mathrm{Dg}_i \cup \Delta$ and $\mathrm{Dg}_j \cup \Delta$ instead of between $\mathrm{Dg}_i$ and $\mathrm{Dg}_j$. Carriere et al. [2017], for instance, used Wasserstein distance between $\mathrm{Dg}_i$ and $\mathrm{Dg}_j$ where its transportation plans operate between $\mathrm{Dg}_i \cup \mathrm{Dg}_{j\Delta}$ and $\mathrm{Dg}_j \cup \mathrm{Dg}_{i\Delta}$ (nonnegative, not necessarily normalized measures with same masses). Here, we denote $\mathrm{Dg}_{i\Delta} := \{ \Pi_\Delta(u) \mid u \in \mathrm{Dg}_i \}$ where $\Pi_\Delta(u)$ is a projection of a point $u$ on the diagonal set $\Delta$. Following this line of work, we also consider a distance between two measures $\mathrm{Dg}_i \cup \mathrm{Dg}_{j\Delta}$ and $\mathrm{Dg}_i \cup \mathrm{Dg}_{j\Delta}$ as a distance between $\mathrm{Dg}_i$ and $\mathrm{Dg}_j$ for the Fisher information metric.

**Definition 1.** *Let $Dg_i, Dg_j$ be two finite and bounded persistence diagrams. The Fisher information metric between $Dg_i$ and $Dg_j$ is defined as follows,*

$$d_{FIM}(Dg_i, Dg_j) := d_{\mathcal{P}} \left( \rho_{(Dg_i \cup Dg_{j\Delta})}, \rho_{(Dg_j \cup Dg_{j\Delta})} \right). \tag{3}$$

**Lemma 3.1.** *Let $\mathbb{D}$ be the set of bounded and finite persistent diagrams. Then, $(d_{FIM} - \tau)$ is negative definite on $\mathbb{D}$ for all $\tau \geq \frac{\pi}{2}$.*

*Proof.* Let consider the function $\tau - \arccos(\xi)$ where $\tau \geq \frac{\pi}{2}$ and $\xi \in [0, 1]$, then apply the Taylor series expansion for $\arccos(\xi)$ at 0, we have

$$\tau - \arccos(\xi) = \tau - \frac{\pi}{2} + \sum_{i=0}^\infty \frac{(2i)!}{2^{2i}(i!)^2(2i+1)} x^{2i+1}.$$

So, all coefficients of the Taylor series expansion are nonnegative. Following [Schoenberg, 1942] (Theorem 2, p. 102), for $\tau \geq \frac{\pi}{2}$ and $\xi \in [0,1]$, $\tau - \arccos(\xi)$ is positive definite. Consequently, $\arccos(\xi) - \tau$ is negative definite. Furthermore, for any PDs $\mathrm{Dg}_i$ and $\mathrm{Dg}_j$ in $\mathbb{D}$, we have

$$0 \leq \int \sqrt{\bar{\rho}_i(x)\bar{\rho}_j(x)}\mathrm{d}x \leq 1,$$

where we denote $\bar{\rho}_i = \rho_{(\mathrm{Dg}_i \cup \mathrm{Dg}_{j\Delta})}$ and $\bar{\rho}_j = \rho_{(\mathrm{Dg}_j \cup \mathrm{Dg}_{i\Delta})}$. The lower bound is due to nonnegativity of the probability simplex while the upper bound follows from the Cauchy-Schwarz inequality. Hence, $d_{\mathtt{FIM}} - \tau$ is negative definite on $\mathbb{D}$ for all $\tau \geq \frac{\pi}{2}$. ∎

Based on Lemma 3.1, we propose a positive definite kernel for PDs under the Fisher information geometry by following [Berg et al., 1984] (Theorem 3.2.2, p.74), namely the Persistence Fisher kernel,

$$k_{\mathrm{PF}}(\mathrm{Dg}_i, \mathrm{Dg}_j) := \exp\left(-t d_{\mathtt{FIM}}(\mathrm{Dg}_i, \mathrm{Dg}_j)\right), \tag{4}$$

where $t$ is a positive scalar since we can rewrite the Persistence Fisher kernel as $k_{\mathrm{PF}}(\mathrm{Dg}_i, \mathrm{Dg}_j) = \alpha \exp\left(-t\left(d_{\mathtt{FIM}}(\mathrm{Dg}_i, \mathrm{Dg}_j) - \tau\right)\right)$ where $\tau \geq \frac{\pi}{2}$ and $\alpha = \exp\left(-t\tau\right) > 0$.

To the best of our knowledge, the $k_{\mathrm{PF}}$ is the first kernel relying on the Fisher information geometry for measuring the similarity of PDs. Moreover, the $k_{\mathrm{PF}}$ is positive definite *without any approximation*.

**Remark 1.** *Let $\mathbb{S}_+ := \left\{\nu \mid \int \nu^2(x)\mathrm{d}x = 1, \nu(x) \geq 0\right\}$ be the positive orthant of the sphere, and define the Hellinger mapping $h(\cdot) := \sqrt{\cdot}$, where the square root is an element-wise function which transforms the probability simplex $\mathbb{P}$ into $\mathbb{S}_+$. The Fisher information metric between $\rho_i$ and $\rho_j$ in $\mathbb{P}$ (Equation (2)) is equivalent to the geodesic distance between $h(\rho_i)$ and $h(\rho_j)$ in $\mathbb{S}_+$. From [Levy and Loeve, 1965], the geodesic distance in $\mathbb{S}_+$ is a measure definite kernel distance. Following [Istas, 2012] (Proposition 2.8), the geodesic distance in $\mathbb{S}_+$ is negative definite. This result is also noted in [Feragen et al., 2015]. From [Berg et al., 1984] (Theorem 3.2.2, p.74), the Persistence Fisher kernel is positive definite. Therefore, our proof technique is not only independent and direct for the Fisher information metric on the probability simplex without relying on the geodesic distance on $\mathbb{S}_+$, but also valid for the case of infinite dimensions due to [Schoenberg, 1942] (Theorem 2, p. 102).*

**Remark 2.** *A closely related kernel to the Persistence Fisher kernel is the diffusion kernel [Lafferty and Lebanon, 2005] (p. 140), based on the heat equation on the Riemannian manifold defined by the Fisher information metric to exploit the geometric structure of statistical manifolds. A generalized family of kernels for the diffusion kernel is exploited in [Jayasumana et al., 2015, Feragen et al., 2015]. To the best of our knowledge, the diffusion kernel has not been used for measuring the similarity of PDs. If one uses the Fisher information metric (Definition 1) for PDs, and then plug the distance into the diffusion kernel, one obtains a similar form to our proposed Persistence Fisher kernel. A slight difference is that the diffusion kernel relies on $d_{FIM}^2$ while the Persistence Fisher kernel is built upon $d_{FIM}$ itself. However, the Persistence Fisher kernel is positive definite while it is unclear whether the diffusion kernel is positive definite[3].*

**Computation.** Given two finite PDs $\mathrm{Dg}_i$ and $\mathrm{Dg}_j$ with cardinalities bounded by $N$, in practice, we consider a finite set $\Theta := \mathrm{Dg}_i \cup \mathrm{Dg}_{j\Delta} \cup \mathrm{Dg}_j \cup \mathrm{Dg}_{i\Delta}$ without multiplicity in $\mathbb{R}^2$ for smoothed and normalized measures $\rho_{(\cdot)}$ (Equation 1)[4]. Then, let $m$ be the cardinality of $\Theta$, we have $m \leq 4N$. Consequently, the time complexity of $\rho_{(\cdot)}$ is $O(Nm)$. For acceleration, we propose to apply the Fast Gauss Transform [Greengard and Strain, 1991, Morariu et al., 2009] to approximate the sum of Gaussian functions in $\rho_{(\cdot)}$ with a bounded error. The time complexity of $\rho_{(\cdot)}$ is reduced from $O(Nm)$ to $O(N + m)$. Due to the low dimension of points in PDs ($\mathbb{R}^2$), this approximation by the Fast Gauss Transform is very efficient in practice. Additionally, $d_{\mathcal{P}}$ (Equation (2)) is evaluated between two points in the $m$-dimensional probability simplex $\mathbb{P}_{m-1}$ where $\mathbb{P}_{m-1} := \left\{x \mid x \in \mathbb{R}_+^m, \|x\|_1 = 1\right\}$. So, the time complexity of the Persistence Fisher kernel $k_{\mathrm{PF}}$ between two smoothed and normalized measures is $O(m)$. Hence, the time complexity of $k_{\mathrm{PF}}$ between $\mathrm{Dg}_i$ and $\mathrm{Dg}_j$ is $O(N^2)$, or $O(N)$ for the acceleration version with Fast Gauss Transform. We summarize the computation of $d_{\mathtt{FIM}}$ in Algorithm

Table 1: A comparison for time complexities and metric preservation of kernels for PDs. Noted that the SW kernel is built upon the SW distance (an *approximation* of Wasserstein metric) while the PF kernel uses the Fisher information metric *without approximation*.

| | $k_{\text{PSS}}$ | $k_{\text{PWG}}$ | $k_{\text{SW}}$ | $k_{\text{PF}}$ |
|---|---|---|---|---|
| Time complexity | $O(N^2)$ | $O(N^2)$ | $O(N^2 \log N)$ | $O(N^2)$ |
| Time complexity with approximation | $O(N)$ | $O(N)$ | $O(MN \log N)$ | $O(N)$ |
| Metric preservation | | | ✓ | ✓ |

1, where the second and third steps can be approximated with a bounded error via Fast Gaussian Transform with a linear time complexity $O(N)$. Source code for Algorithm 1 can be obtained in http://github.com/lttam/PersistenceFisher. We recall that the time complexity of the Wasserstein distance between $\text{Dg}_i$ and $\text{Dg}_j$ is $O(N^3 \log N)$ [Pele and Werman, 2009] (§2.1). For the Sliced Wasserstein distance (an approximation of Wasserstein distance), the time complexity is $O(N^2 \log N)$ [Carriere et al., 2017], or $O(MN \log N)$ for its approximation with $M$ projections [Carriere et al., 2017]. We also summary a comparison for the time complexity and metric preservation of $k_{\text{PF}}$ and related kernels for PDs in Table 1.

---

**Algorithm 1** Compute $d_{\texttt{FIM}}$ for persistence diagrams

---

**Input:** Persistence diagrams $\text{Dg}_i$, $\text{Dg}_j$, and a bandwith $\sigma > 0$ for smoothing
**Output:** $d_{\texttt{FIM}}$
1: Let $\Theta \leftarrow \text{Dg}_i \cup \text{Dg}_{j\Delta} \cup \text{Dg}_j \cup \text{Dg}_{i\Delta}$ (a set for smoothed and normalized measures)
2: Compute $\bar{\rho}_i = \rho_{\left(\text{Dg}_i \cup \text{Dg}_{j\Delta}\right)} \leftarrow \left[ \frac{1}{Z} \sum_{u \in \text{Dg}_i \cup \text{Dg}_{j\Delta}} \texttt{N}(x; u, \sigma I) \right]_{x \in \Theta}$
$\qquad$ where $Z \leftarrow \sum_{x \in \Theta} \sum_{u \in \text{Dg}_i \cup \text{Dg}_{j\Delta}} \texttt{N}(x; u, \sigma I)$
3: Compute $\bar{\rho}_j = \rho_{\left(\text{Dg}_j \cup \text{Dg}_{i\Delta}\right)}$ similarly as $\bar{\rho}_i$.
4: Compute $d_{\texttt{FIM}} \leftarrow \arccos\left(\left\langle \sqrt{\bar{\rho}_i}, \sqrt{\bar{\rho}_j} \right\rangle\right)$ where $\langle \cdot, \cdot \rangle$ is a dot product and $\sqrt{\cdot}$ is element-wise.

---

## 4 Theoretical Analysis

In this section, we analyze for the Persistence Fisher kernel $k_{\text{PF}}$ (in Equation (4)) where the Hellinger mapping $h$ of a smoothed and normalized measure $\rho_{(\cdot)}$ is on the positive orthant of the $d$-dimension unit sphere $\mathbb{S}_{d-1}^+$ where $\mathbb{S}_{d-1}^+ := \left\{ x \mid x \in \mathbb{R}_+^d, \|x\|_2 = 1 \right\}$[5]. Let $\text{Dg}_i, \text{Dg}_j$ be PDs in the set $\mathbb{D}$ of bounded and finite PDs, and $\mu$ be the uniform probability distribution on $\mathbb{S}_{d-1}^+$. We denote $x_i$ and $x_j \in \mathbb{S}_{d-1}^+$ as corresponding mapped points through the Hellinger mapping $h$ of smoothed and normalized measures $\rho_{(\text{Dg}_i \cup \text{Dg}_{j\Delta})}$ and $\rho_{(\text{Dg}_j \cup \text{Dg}_{i\Delta})}$ respectively. Then, we rewrite the Persistence Fisher kernel between $x_i$ and $x_j$ as follows,

$$k_{\text{PF}}(x_i, x_j) = \exp\left(-t \arccos\left(\langle x_i, x_j \rangle\right)\right). \tag{5}$$

**Eigensystem.** Let $T_{k_{\text{PF}}} : L_2(\mathbb{S}_{d-1}^+, \mu) \rightarrow L_2(\mathbb{S}_{d-1}^+, \mu)$ be the integral operator induced by the Persistence Fisher kernel $k_{\text{PF}}$, which is defined as

$$\left(T_{k_{\text{PF}}} f\right)(\cdot) := \int k_{\text{PF}}(x, \cdot) f(x) \mathrm{d}\mu(x).$$

Following [Smola et al., 2001] (Lemma 4), we derive an eigensystem of the integral operator $T_{k_{\text{PF}}}$ as in Proposition 1.

**Proposition 1.** *Let $\{a_i\}_{i \geq 0}$ be the coefficients of Legendre polynomial expansion of the Persistence Fisher kernel $k_{PF}(x, z)$ defined on $\mathbb{S}_{d-1}^+ \times \mathbb{S}_{d-1}^+$ as in Equation (5),*

$$k_{PF}(x, z) = \sum_{i=0}^{\infty} a_i P_i^d(\langle x, z \rangle), \tag{6}$$

where $P_i^d$ is the associated Legendre polynomial of degree $i$. Let $|\mathbb{S}_{d-1}| := \frac{2\pi^{d/2}}{\Gamma(d/2)}$ denote the surface of $\mathbb{S}_{d-1}$ where $\Gamma(\cdot)$ is the Gamma function, $N(d,i) := \frac{(d+2i-2)(d+i-3)!}{(d-2)!i!}$ denote the multiplicity of spherical harmonics of order $i$ on $\mathbb{S}_{d-1}$, and $\left\{Y_{i,j}^d\right\}_{1 \le j \le N(d,i)}$ denote any fixed orthonormal basis for the subspace of all homogeneous harmonics of order $i$ on $\mathbb{S}_{d-1}$. Then, the eigensystem $(\lambda_{i,j}, \phi_{i,j})$ of the integral operator $T_{k_{PF}}$ induced by the Persistence Fisher kernel $k_{PF}$ is

$$\phi_{i,j} = Y_{i,j}^d, \tag{7}$$

$$\lambda_{i,j} = \frac{a_i |\mathbb{S}_{d-1}|}{N(d,i)} \tag{8}$$

of multiplicity $N(d,i)$.

*Proof.* From the Addition Theorem [Muller, 2012] (Theorem 2, p. 18) and the Funk-Hecke formula [Muller, 2012] (§4, p. 29), we have $\sum_{j=1}^{N(d,i)} Y_{i,j}^d(x)Y_{i,j}^d(z) = \frac{N(d,i)}{|\mathbb{S}_{d-1}|}P_i^d(\langle x, z \rangle)$, then replace $P_i^d$ into Equation (6), and note that $\int_{\mathbb{S}_{d-1}} Y_{i,j}^d(x)Y_{i',j'}^d(x)\mathrm{d}x = \delta_{i,i'}\delta_{j,j'}$, we complete the proof. ∎

**Proposition 2.** *All coefficients of Legendre polynomial expansion of the Persistence Fisher kernel are nonnegative.*

*Proof.* From Lemma 3.1, the $k_{PF}$ is positive definite. Applying Schoenberg [1942] (Theorem 1, p. 101) for $k_{PF}$ defined on $\mathbb{S}_{d-1}^+ \times \mathbb{S}_{d-1}^+$ as in Equation (5), we obtain the result. ∎

The eigensystem of the integral operator $T_{k_{PF}}$ induced by the PF kernel plays an important role to derive generalization error bounds for kernel machines with the proposed PF kernel via covering numbers and Rademacher averages as in Proposition 3 and Proposition 4 respectively.

**Covering numbers.** Given a set of finite points $\mathtt{S} = \left\{x_i \mid x_i \in \mathbb{S}_{d-1}^+, d \ge 3\right\}$, the Persistence Fisher kernel hypothesis class with $R$-bounded weight vectors for $\mathtt{S}$ is defined as follows

$$\mathcal{F}_R(\mathtt{S}) = \left\{\mathtt{f} \mid \mathtt{f}(x_i) = \langle w, \phi(x_i) \rangle_{\mathcal{H}}, \|w\|_{\mathcal{H}} \le R\right\},$$

where $\langle \phi(x_i), \phi(x_j) \rangle_{\mathcal{H}} = k_{PF}(x_i, x_j)$. $\langle \cdot, \cdot \rangle_{\mathcal{H}}$ and $\|\cdot\|_{\mathcal{H}}$ are an inner product and a norm in the corresponding Hilbert space respectively. Following [Guo et al., 1999], we derive bounds on the generalization performance of the PF kernel on kernel machines via the covering numbers $\mathcal{N}(\cdot, \mathcal{F}_R(\mathtt{S}))$ [Shalev-Shwartz and Ben-David, 2014] (Definition 27.1, p. 337) as in Proposition 3.

**Proposition 3.** *Assume the number of non-zero coefficients $\{a_i\}$ in Equation (6) is finite, and $r$ is the maximum index of the non-zero coefficients. Let $q := \arg\max_i \lambda_{i,\cdot}$, choose $\alpha \in \mathbb{N}$ such that $\alpha < \left(\frac{\lambda_{q,\cdot}}{\lambda_{i,\cdot}}\right)^{\frac{N(d,q)}{2}}$ with $i \ne q$, and define $\varepsilon := 6R\sqrt{N(d,r)\left(a_q\alpha^{-2/N(d,q)} + \sum_{i=0,i\ne q}^{\infty} a_i\right)}$. Then,*

$$\sup_{x_i \in \mathtt{S}} \mathcal{N}(\varepsilon, \mathcal{F}_R(\mathtt{S})) \le \alpha.$$

*Proof.* From [Minh et al., 2006] (Lemma 3), we have $\left\|Y_{i,j}^d\right\|_\infty \le \sqrt{\frac{N(d,i)}{|\mathbb{S}_{d-1}|}}$. It is easy to check that $\forall d \ge 3, i \ge j \ge 0$, we have $N(d,i) \ge N(d,j)$. Therefore, following Proposition 1, all eigenfunctions of $k_{PF}$ satisfy that $\left\|Y_{i,j}^d\right\|_\infty \le \sqrt{\frac{N(d,r)}{|\mathbb{S}_{d-1}|}}$. Additionally, the multiplicity of $\lambda_{i,\cdot}$ is $N(d,i)$, and $N(d,i)\lambda_{i,\cdot} = a_i |\mathbb{S}_{d-1}|$ (Equation (8)). Hence, from [Guo et al., 1999] (Theorem 1), we obtain the result. ∎

**Rademacher averages.** We provide a different family of generalization error bounds via Rademacher averages [Bartlett et al., 2005]. By plugging the eigensystem of the PF kernel as in Proposition 1 into the localized averages of function classes based on the PF kernel with respect to the uniform probability distribution $\mu$ on $\mathbb{S}_{d-1}^+$ [Mendelson, 2003] (Theorem 2.1), we obtain a bound as in Proposition 4.

**Proposition 4.** *Let $\{x_i\}_{1 \le i \le m}$ be independent, distributed according to the uniform probability distribution $\mu$ on $\mathbb{S}_{d-1}^+$, denote $\{\sigma_i\}_{1 \le i \le m}$ for independent Rademacher random variables, $\mathcal{H}_{k_{PF}}$ for the unit ball of the reproducing kernel Hilbert space corresponding with the Riemanian manifold kernel $k_{PF}$, and let $q := \arg\max_i \lambda_{i,\cdot}$. If $\lambda_{q,\cdot} \ge 1/m$, for $\tau \ge 1/(m\,|\mathbb{S}_{d-1}|)$,*

*let $\Psi(\tau) := \sqrt{ |\mathbb{S}_{d-1}| \left( \displaystyle\sum_{a_i < \tau N(d,i)} a_i + \tau \sum_{a_i \ge \tau N(d,i)} N(d,i) \right)}$, then there are absolute constants $C_\ell$ and $C_u$*

*which satisfy*

$$C_\ell \Psi(\tau) \le \mathbb{E} \sup_{\substack{\mathtt{f} \in \mathcal{H}_{k_{PF}} \\ \frac{\mathbb{E}_\mu \mathtt{f}^2}{|\mathbb{S}_{d-1}|} \le \tau}} \left| \sum_{i=1}^{m} \sigma_i \mathtt{f}(x_i) \right| \le C_u \Psi(\tau), \tag{9}$$

*where $\mathbb{E}$ is an expectation.*

From Proposition 3 and Proposition 4, a decay rate of the eigenvalues of the integral operator $T_{k_{PF}}$ is relative with the capacity of the kernel learning machines. When the decay rate of the eigenvalues is large, the capacity of kernel machines is reduced. So, if the training error of kernel machines is small, then it can lead to better bounds on generalization error. The resulting bounds for both the covering number (Proposition 3) and the Rademacher averages (Proposition 4) are essentially the same as the standard ones for a Gaussian kernel on a Euclidean space.

**Bounding for $k_{\mathbf{PF}}$ induced squared distance with respect to $d_{\mathtt{FIM}}$.** The squared distance induced by the PF kernel, denoted as $d_{k_{PF}}^2$, can be computed by the Hilbert norm of the difference between two corresponding mappings. Given two persistent diagram $\mathrm{Dg}_i$ and $\mathrm{Dg}_j$, we have

$$d_{k_{PF}}^2 \left( \mathrm{Dg}_i, \mathrm{Dg}_j \right) := k_{\mathrm{PF}} \left( \mathrm{Dg}_i, \mathrm{Dg}_i \right) + k_{\mathrm{PF}} \left( \mathrm{Dg}_j, \mathrm{Dg}_j \right) - 2k_{\mathrm{PF}} \left( \mathrm{Dg}_i, \mathrm{Dg}_j \right).$$

We recall that $k_{PF}$ is based on the Fisher information geometry. So, it is of interest to bound the PF kernel induced squared distance $d_{k_{PF}}^2$ with respect to the corresponding Fisher information metric $d_{\mathtt{FIM}}$ between PDs as in Lemma 4.1.

**Lemma 4.1.** *Let $\mathbb{D}$ be the set of bounded and finite persistent diagrams. Then, $\forall Dg_i, Dg_j \in X$,*

$$d_{k_{PF}}^2(Dg_i, Dg_j) \le 2t d_{FIM}(Dg_i, Dg_j),$$

*where $t$ is a parameter of $k_{PF}$.*

*Proof.* We have $d_{k_{PF}}^2(\mathrm{Dg}_i, \mathrm{Dg}_j) = 2\left(1 - k_{\mathrm{PF}}\left(\mathrm{Dg}_i, \mathrm{Dg}_j\right)\right) = 2(1 - \exp\left(-t d_{\mathtt{FIM}}\left(\mathrm{Dg}_i, \mathrm{Dg}_j\right)\right) \le 2t d_{\mathtt{FIM}}\left(\mathrm{Dg}_i, \mathrm{Dg}_j\right)$, since $1 - \exp(-a) \le a, \forall a \ge 0$. ∎

From Lemma 4.1, it implies that the Persistence Fisher kernel is stable on Riemannian geometry in a similar sense as the work of Kwitt et al. [2015], and Reininghaus et al. [2015] on Wasserstein geometry.

**Infinite divisibility for the Persistence Fisher kernel.**

**Lemma 4.2.** *The Persistence Fisher kernel $k_{PF}$ is infinitely divisible.*

*Proof.* For $\mathtt{m} \in \mathbb{N}^*$, let $k_{\mathrm{PF_m}} := \exp\left(-\frac{t}{\mathtt{m}} d_{\mathtt{FIM}}\right)$, so $\left(k_{\mathrm{PF_m}}\right)^{\mathtt{m}} = k_{\mathrm{PF}}$ and note that $k_{\mathrm{PF_m}}$ is positive definite. Hence, following Berg et al. [1984] (§3, Definition 2.6, p. 76), we have the result. ∎

As for infinitely divisible kernels, the Gram matrix of the PF kernel does not need to be recomputed for each choice of $t$ (Equation (4)), since it suffices to compute the Fisher information metric between PDs in training set only once. This property is shared with the Sliced Wasserstein kernel [Carriere et al., 2017]. However, neither Persistence Scale Space kernel [Reininghaus et al., 2015] nor Persistence Weighted Gaussian kernel [Kusano et al., 2016] has this property.

Table 2: Results on SVM classification. The averaged accuracy (%) and standard deviation are shown.

|  | MPEG7 | Orbit |
|---|---|---|
| $k_{\text{PSS}}$ | $73.33 \pm 4.17$ | $72.38 \pm 2.41$ |
| $k_{\text{PWG}}$ | $74.83 \pm 4.36$ | $76.63 \pm 0.66$ |
| $k_{\text{SW}}$ | $76.83 \pm 3.75$ | $83.60 \pm 0.87$ |
| Prob+$k_G$ | $55.83 \pm 5.45$ | $72.89 \pm 0.62$ |
| Tang+$k_G$ | $66.17 \pm 4.01$ | $77.32 \pm 0.72$ |
| $\boldsymbol{k_{\text{PF}}}$ | $\mathbf{80.00 \pm 4.08}$ | $\mathbf{85.87 \pm 0.77}$ |

# 5    Experimental Results

We evaluated the Persistence Fisher kernel with support vector machines (SVM) on many benchmark datasets. We consider five baselines as follows: (i) the Persistence Scale Space kernel ($k_{\text{PSS}}$), (ii) the Persistence Weighted Gaussian kernel ($k_{\text{PWG}}$), (iii) the Sliced Wasserstein kernel ($k_{\text{SW}}$), (iv) the smoothed and normalized measures in the probability simplex with the Gaussian kernel (Prob + $k_G$), and (v) the tangent vector representation [Anirudh et al., 2016] with the Gaussian kernel (Tang + $k_G$). Practically, Euclidean metric is not a suitable geometry for the probability simplex [Le and Cuturi, 2015a,b]. So, the (Prob + $k_G$) approach may not work well for PDs. For hyper-parameters, we typically choose them through cross validation. For baseline kernels, we follow their corresponding authors to form sets of hyper-parameter candidates, and the bandwidth of the Gaussian kernel in (Prob + $k_G$) and (Tang + $k_G$) is chosen from $10^{\{-3:1:3\}}$. For the Persistence Fisher kernel, there are 2 hyper-parameters: $t$ (Equation (4)) and $\sigma$ for smoothing measures (Equation (1)). We choose $1/t$ from $\{q_1, q_2, q_5, q_{10}, q_{20}, q_{50}\}$ where $q_s$ is the $s\%$ quantile of a subset of Fisher information metric between PDs, observed on the training set, and $\sigma$ from $\{10^{-3:1:3}\}$. For SVM, we use Libsvm (one-vs-one) [Chang and Lin, 2011] for multi-class classification, and choose a regularization parameter of SVM from $\{10^{-2:1:2}\}$. For PDs, we used the DIPHA toolbox[6].

## 5.1    Orbit Recognition

It is a synthesized dataset proposed by [Adams et al., 2017] (§6.4.1) for *linked twist map* which is a discrete dynamical system modeling flow. The linked twist map is used to model flows in DNA microarrays [Hertzsch et al., 2007]. Given a parameter $r > 0$, and initial positions $(s_0, t_0) \in [0,1]^2$, its orbit is described as $s_{i+1} = s_i + rt_i(1 - t_i) \mod 1$, and $t_{i+1} = t_i + rs_{i+1}(1 - s_{i+1}) \mod 1$. Adams et al.

Table 3: Computational time (seconds) with approximation. For each dataset, the first number in the parenthesis is the number of PDs while the second one is the maximum number of points in PDs.

|  | Orbit (5K/300) | MPEG7 (200/80) | Granular (35/20.4K) | SiO$_2$ (80/30K) |
|---|---|---|---|---|
| $k_{\text{SW}}$ | 6473 | 1.55 | 8.30 | 249 |
| $k_{\text{PWG}}$ | 8756 | 5.23 | 17.44 | 288 |
| $k_{\text{PSS}}$ | 11024 | 7.51 | 38.14 | 515 |
| $\boldsymbol{k_{\text{PF}}}$ | **9891** | **6.63** | **22.70** | **318** |

[2017] proposed 5 classes, corresponding to 5 different parameters $r = 2.5, 3.5, 4, 4.1, 4.3$. For each parameter $r$, we generated 1000 orbits where each orbit has 1000 points with random initial positions. We randomly split 70%/30% for training and test, and repeated 100 times. We extract only 1-dimensional topological features with Vietoris-Rips complex filtration [Edelsbrunner and Harer, 2008] for PDs. The accuracy results on SVM are summarized in the third column of Table 2. The PF kernel outperforms all other baselines. The (Prob + $k_G$) does not performance well as expected. Moreover, the $k_{\text{PF}}$ and $k_{\text{SW}}$ which enjoy the Fisher information geometry and Wasserstein geometry for PDs respectively, clearly outperform other approaches. As in the second column of Table 3, the computational time of $k_{\text{PF}}$ is faster than $k_{\text{PSS}}$, but slower than $k_{\text{SW}}$ and $k_{\text{PWG}}$ for PDs.

## 5.2    Object Shape Classification

We consider a 10-class subset[7] of MPEG7 object shape dataset [Latecki et al., 2000]. Each class has 20 samples. We resize each image such that its length is shorter or equal 256, and extract a boundary for object shapes before computing PDs. For simplicity, we only consider 1-dimensional topological

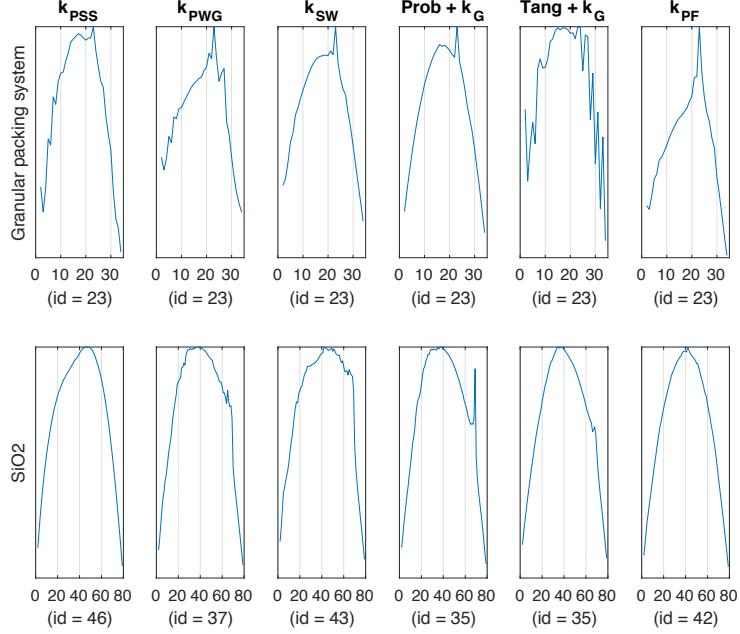

Figure 2: The kernel Fisher discriminant ratio (KFDR) graphs.

features with the traditional Vietoris-Rips complex filtration [Edelsbrunner and Harer, 2008] for PDs[8]. We also randomly split $70\%/30\%$ for training and test, and repeated 100 times. The accuracy results on SVM are summarized in the second column of Table 2. The Persistence Fisher kernel compares favorably with other baseline kernels for PDs. All approaches based on the implicit representation via kernels for PDs outperform ones based on the explicit vector representation with Gaussian kernel by a large margin. Additionally, the $k_{PF}$ and $k_{SW}$ also compares favorably with other approaches. As in the third column of Table 3, the computational time of $k_{PF}$ is comparative with $k_{PWG}$ and $k_{PSS}$, but slower than the $k_{SW}$.

## 5.3 Change Point Detection for Material Data Analysis

We evaluated the proposed kernel for the change point detection problem for material data analysis on granular packing system [Francois et al., 2013] and $SiO_2$[Nakamura et al., 2015] datasets. We use the kernel Fisher discriminant ratio [Harchaoui et al., 2009] (KFDR) as a statistical quantity and set $10^{-3}$ for the regularization of KFDR as in [Kusano et al., 2018]. We use the ball model filtration to extract the 2-dimensional topological features of PDs for granular packing system dataset, and 1-dimensional topological features of PDs for $SiO_2$ dataset. We illustrate the KFDR graphs for the granular packing system and $SiO_2$ datasets in Figure 2. For granular tracking system dataset, all methods obtain the change point as the $23^{rd}$ index. They supports the observation result in [Anonymous, 1972] (corresponding id = 23). For the $SiO_2$ datasets, all methods obtain the results within the supported range ($35 \leq$ id $\leq 50$) from the traditional physical approach [Elliott, 1983]. The $k_{PF}$ compares favorably with other baseline approaches as in Figure 2. As in the fourth and fifth columns of Table 3, $k_{PF}$ is faster than $k_{PSS}$, but slower than $k_{SW}$ and $k_{PWG}$.

## 6 Conclusions

In this work, we propose the positive definite Persistence Fisher (PF) kernel for persistence diagrams (PDs). The PF kernel is relied on the Fisher information geometry *without approximation* for PDs. Moreover, the proposed kernel has many nice properties from both theoretical and practical aspects such as stability, infinite divisibility, linear time complexity over the number of points in PDs, and improving performances of other baseline kernels for PDs as well as implicit vector representation with Gaussian kernel for PDs in many different tasks on various benchmark datasets.

**Acknowledgments**

We thank Ha Quang Minh, and anonymous reviewers for their comments. TL acknowledges the support of JSPS KAKENHI Grant number 17K12745. MY was supported by the JST PRESTO program JPMJPR165A.

## Footnotes

[1] In case, $\Theta$ is an *infinite* set, then the corresponding probability simplex $\mathbb{P}$ has *infinite* dimensions.

[2] FIM is also known as a particular pull-back metric on Riemannian manifold [Le and Cuturi, 2015b].

[3]Although the heat kernel is positive definite, the diffusion kernel on the probability simplex—the heat kernel on multinomial manifold—does not have an explicit form. In practice, the diffusion kernel equation [Lafferty and Lebanon, 2005] (p. 140) is only its first-order approximation.

[4]We leave the computation with an *infinite* set $\Theta$ for future work.

[5]It is corresponding to a finite set $\Theta$.

[6]https://github.com/DIPHA/dipha

[7]The 10-classes are: apple, bell, bottle, car, classic, cup, device0, face, Heart and key.

[8]A more advanced filtration for this task was proposed in [Turner et al., 2014].

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
