[Supplementary Material]

# *Supplementary Material for:* Persistence Fisher Kernel: A Riemannian Manifold Kernel for Persistence Diagrams

**Tam Le**
RIKEN Center for Advanced Intelligence Project, Japan
`tam.le@riken.jp`

**Makoto Yamada**
Kyoto University, Japan
RIKEN Center for Advanced Intelligence Project, Japan
`makoto.yamada@riken.jp`

## 1 Some Traditional Filtrations for Persistence Diagrams

Figure 1: An illustration for persistence diagrams with some popular filtrations. (a) A set of points as an input. (b) A ball model filtration. (c) Cech complex filtration. (d) Vietoris-Rips complex filtration (it has only 1 ring since it contains a 2-simplex, illustrated as the orange triangle). (e) An illustration of a birth of a ring in the ball model filtration. (f) An illustration of a death for a ring in the ball model filtration. (g) A sub-level set filtration (a connected component has a birth at $\mathfrak{F}_p$, and a death at $\mathfrak{F}_q$). In this illustration, both the ball model filtration and Cech complex filtration have 2 rings, but there is only 1 ring for Vietoris-Rips complex filtration due to the 2-simplex. For sub-level set filtration, there are 2 connected components $(p, q)$ and $(t, \infty)$. Hence, the persistence diagram of 0-dimension topological feature is that $\text{Dg} = \{(p, q); (t, \infty)\}$.

We provide some traditional filtrations to illustrate persistence diagrams as follows,

**Ball model filtration.** Let $X = \{x_1, x_2, ..., x_m\}$ be a finite set in a metric space as in Figure 1 (a), and $B(x, a)$ be a ball with a center $x$ and a radius $a$. We denote $X_a := \cup_{x_i \in X} B(x_i, a)$ for $a \geq 0$. For $a < 0$, we define $X_a := \emptyset$. Therefore, $\{X_a \mid a \in \mathbb{R}\}$ can be used as a filtration, illustrated in

Figure 1 (b). For example, Figure 1 (e) shows a birth for a ring at $X_p$ while Figure 1 (f) illustrates that the ring is death at $X_q$. Therefore, a point $(p, q)$ is in the persistence diagram of 1-dimensional topological feature for the set $X$.

**Cech complex filtration.**  Given a set $X = \{x_1, x_2, ..., x_m\}$ in a metric space $(T, d_T)$. For $a \geq 0$, we form a $t$-simplex from a $(t + 1)$-point subset $X_{t+1}$ of $X$ if there exist $x' \in M$, such that $d_T(x, x') \leq a, \forall x \in X_{t+1}$. The set of all these simplices is called the Cech complex of $X$ with parameter $a \geq 0$, denoted as $\mathfrak{C}(X, a)$. For $a < 0$, we definite $\mathfrak{C}(X, a) := \emptyset$. Therefore, $\{\mathfrak{C}(X, a) \mid a \in \mathbb{R}\}$ can be considered as a filtration and illustrated in Figure 1 (c). When $T \subset \mathbb{R}^q$, the topology of $\mathfrak{C}(X, a)$ is homotopy equivalent to $X_a$ [Hatcher, 2002] (p. 257). Consequently, the persistence diagrams with Cech complex filtration equals to the persistence diagrams with ball model filtration.

**Vietoris-Rips complex (a.k.a. Rips complex) filtration.**  Given a set $X = \{x_1, x_2, ..., x_m\}$ in a metric space $(T, d_T)$. For $a \geq 0$, we form a $t$-simplex from a $(t + 1)$-point subset $X_{t+1}$ of $X$ which satisfies $d_T(x, z) \leq 2a, \forall x, z \in X_{t+1}$. The set of all these simplices is called Vietoris-Rips complex of $X$ with parameter $a \geq 0$, denoted as $\mathfrak{R}(X, a)$. For $a < 0$, we define $\mathfrak{R}(X, a) = \emptyset$. Therefore, $\{\mathfrak{R}(X, a) \mid a \in \mathbb{R}\}$ can be used as a filtration as illustrated in Figure 1 (d).

**Sub-level set filtration.**  Let $T$ be a topological space, given a function $\mathfrak{f} : T \to \mathbb{R}$ as an input, and defined a sub-level set $\mathfrak{F}_a := \mathfrak{f}^{-1}((-\infty, a])$. Thus, $\{\mathfrak{F}_a \mid a \in \mathbb{R}\}$ can be regarded as a filtration as in Figure 1 (g). For example, it is easy to see that a connected component has a birth at $\mathfrak{F}_p$ and it is death at $\mathfrak{F}_q$ as in Figure 1 (g). Thus, a point $(p, q)$ is in persistence diagrams of 0-dimensional topological feature for the given function $\mathfrak{f}$. In Figure 1 (g), persistence diagram of 0-dimensional topological feature for $\mathfrak{f}$ is $\mathrm{Dg} = \{(p, q); (t, \infty)\}$.

## 2 Kernels

We review some important definitions and theorems about kernels.

**Positive definite kernels.**  A function $k : X \times X \to \mathbb{R}$ is called a positive definite kernel if $\forall n \in \mathbb{N}^*, \forall x_1, x_2, ..., x_n \in X, \sum_{i,j} c_i c_j k(x_i, x_j) \geq 0, \forall c_i \in \mathbb{R}$.

**Negative definite kernels.**  A function $k : X \times X \to \mathbb{R}$ is called a negative definite kernel if $\forall n \in \mathbb{N}^*, \forall x_1, x_2, ..., x_n \in X, \sum_{i,j} c_i c_j k(x_i, x_j) \leq 0, \forall c_i \in \mathbb{R}$ such that $\sum_i c_i = 0$.

**Berg-Christensen-Ressel Theorem.**  In [Berg et al., 1984] (Theorem 3.2.2, p.74), if $\kappa$ is a *negative definite* kernel, then $k_t(x, z) := \exp(-t\kappa(x, z))$ is a positive definite kernel for all $t > 0$. For example, Gaussian kernel $k_t(x, z) = \exp\left(-t \|x - z\|_2^2\right)$ is positive definite since it is easy to check that squared Euclidean distance is indeed a negative definite kernel[1].

**Schoenberg Theorem.**  In [Schoenberg, 1942] (Theorem 2, p. 102), a function $f(\langle \cdot, \cdot \rangle)$ defined on the unit sphere in a Hilbert space is positive definite if and only if its Taylor series expansion has only nonnegative coefficients,

$$f(\xi) = \sum_{i=0}^{\infty} a_i \xi^i, \quad \text{with } a_i \geq 0. \tag{1}$$

## 3 Related Kernels for Persistence Diagrams

**Persistence Scale Space kernel ($k_{\text{PSS}}$).**  Reininghaus et al. [2015] proposed the Persistence Scale Space (PSS) kernel, motivated by a heat diffusion problem with a Dirichlet boundary condition. The PSS kernel between two PDs $\mathrm{Dg}_i$ and $\mathrm{Dg}_j$ is defined as $k_{\text{PSS}}(\mathrm{Dg}_i, \mathrm{Dg}_j) :=$

Table 1: Averaged accuracy results (%) on SVM classification. The result of MTF with SVM is cited from [Cang et al., 2015].

|  | MTF | $k_{\text{PSS}}$ | $k_{\text{PWG}}$ | $k_{\text{SW}}$ | Prob+$k_G$ | $k_{\text{PF}}$ |
|---|---|---|---|---|---|---|
| Accuracy (%) | 84.50 | 83.33 | 88.89 | 88.89 | 83.95 | **97.53** |

$\frac{1}{8\pi\sigma} \sum_{\substack{p_i \in \text{Dg}_i \\ p_j \in \text{Dg}_j}} \exp\left(-\frac{\|p_i - p_j\|_2^2}{8\sigma}\right) - \exp\left(-\frac{\|p_i - \bar{p}_j\|_2^2}{8\sigma}\right)$, where $\sigma$ is a scale parameter and if $p = (a, b)$,

then $\bar{p} = (b, a)$, mirrored at the diagonal $\Delta$. The time complexity is $O(N^2)$ where $N$ is the bounded cardinality of PDs. By using the Fast Gauss Transform [Greengard and Strain, 1991] for approximation with bounded error, the time complexity can be reduced to $O(N)$.

**Persistence Weighted Gaussian kernel ($k_{\text{PWG}}$).** Kusano et al. [2016] proposed the Persistence Weighted Gaussian (PWG) kernel by using kernel embedding into the reproducing kernel Hilbert space. Let $k_{G_\sigma}$ be the Gaussian kernel with a positive parameter $\sigma$, and associated reproducing kernel Hilbert space $\mathcal{H}_\sigma$. Let $\mu_i := \sum_{p \in \text{Dg}_i} \arctan(C\text{pers}(p)^q) k_{G_\sigma}(\cdot, p) \in \mathcal{H}_\sigma$, where $C, q$ are positive parameter, and for $p = (a, b)$, a persistence of $p$ is that $\text{pers}(p) := b - a$. Let $\mu_j$ be defined similarly for $\text{Dg}_j$. Given a parameter $\tau > 0$, the persistence weighted Gaussian kernel is defined as $k_{\text{PWG}}(\text{Dg}_i, \text{Dg}_j) := \exp\left(-\frac{\|\mu_i - \mu_j\|_{\mathcal{H}_\sigma}^2}{2\tau^2}\right)$. The time complexity is $O(N^2)$. Furthermore, Kusano et al. [2016] also proposed to use the random Fourier features [Rahimi and Recht, 2008] for computing the Gram matrix of $m$ persistent diagrams with $O(mNu + m^2 u)$ complexity, where $u$ is the number of random variables using for random Fourier features. Thus, the time complexity can be reduced to be linear in $N$.

**Sliced Wasserstein kernel ($k_{\text{SW}}$).** Carriere et al. [2017] proposed the Sliced Wasserstein (SW) kernel, motivated from Wasserstein geometry for PDs. However, it is well-known that the Wasserstein distance is not negative definite. Therefore, it may be necessary to *approximate* the Wasserstein distance to design positive definite kernels on Wasserstein geometry for PDs. Indeed, Carriere et al. [2017] use the SW distance, which is an approximation of Wasserstein distance, for proposing the positive definite SW kernel, defined as $k_{\text{SW}}(\text{Dg}_i, \text{Dg}_j) := \exp\left(-\frac{d_{\text{SW}}(\text{Dg}_i, \text{Dg}_j)}{2\sigma^2}\right)$. The time complexity for the SW distance $d_{\text{SW}}(\text{Dg}_i, \text{Dg}_j)$ is $O(N^2 \log N)$, and for its $M$-projection approximation, it is $O(MN \log N)$.

**Metric preservation.** For those kernel methods for PDs, only the SW kernel preserves the metric between PDs, that is the Wasserstein geometry. Furthermore, Carriere et al. [2017] argued that this property should lead to improve the classification power. In this work, we explore an alternative Riemannian manifold geometry for PDs, namely the Fisher information metric which is also known as a particular pull-back metric on Riemannian manifold [Le and Cuturi, 2015]. Moreover, the proposed positive definite Persistence Fisher kernel is directly built upon the Fisher information metric for PDs *without approximation* while it may be necessary to approximate the Wasserstein distance for designing positive definite kernels on Wasserstein geometry for PDs. Additionally, the time complexity of the Persistence Fisher kernel is also better than the Sliced Wasserstein kernel in term of computation.

## 4 More Experiments on Hemoglobin Classification

We evaluated the Persistence Fisher kernel on Hemoglobin classification for the *taunt* and *relaxed* forms [Cang et al., 2015]. For each form, there are 9 data points, collected by the X-ray crystallography. As in [Kusano et al., 2018], we selected 1 data point from each class for test and used the rest for training. There are totally 81 runs. We also compared with the molecular topological fingerprint (MTF) for SVM [Cang et al., 2015]. We summarize averaged accuracy results on SVM in Table 1. The Persistence Fisher kernel again outperformances other baseline kernels, and also SVM with MTF.

## Footnotes

[1]$\forall n \in \mathbb{N}^*, \forall x_1, x_2, ..., x_n \in X$, and $\forall c_i \in \mathbb{R}$ such that $\sum_i c_i = 0$, we have $\sum_{i,j} c_i c_j \|x_i - x_j\|_2^2 = \sum_i c_i x_i^2 \sum_j c_j + \sum_i c_i \sum_j c_j x_j^2 - 2 \sum_{i,j} c_i c_j x_i x_j = -2 \left(\sum_i c_i x_i\right)^2 \leq 0$.