[Reviews · NeurIPS 2018]

Reviewer 1



This paper proposes a new kernel on persistence diagrams. New kernel has number of appealing properties, the most important is that it is (by definition) psd; this is only interesting in comparison to Wasserstein metric (popular in PD) which does not result in PD "kernels". Paper is well written and clear. The contribution is well defined. Experimental section is solid and fair i.e. compares apples to apples. There is no reason to reject this paper, but it does not seem to be a breakthrough.

Reviewer 2



Update: I had doubts about the smoothing of PD using gaussians, which would normally lead to a continuous distribution even when the set \Theta is finite. However, based on the authors' response, it seems that a different choice of smoothing is used in practice (algorithm 1 , line 2), which is compatible with the results stated in section 3. However, this should be clarified in the paper. The proposed kernel seems to take advantage of the particular geometry of persistence diagrams and leads to better results than more standard kernels (RB, ...). Although it is not a breakthrough and the theory is already well establish, the contribution is relevant, the paper is clearly written and empirical evidence is strong. I vote for accepting the paper. ------------------------------------------------ The paper introduces a new kernel to measure similarity between persistence diagrams. The advantage of such kernel is that it can be computed in linear time (approximately) and is positive definite, unlike other proposed kernels. The paper provides also standard theoretical guarantees for the generalization error. Experimental evidence shows the benefit of using such kernel in terms of generalization error compared to other existing methods. The paper is globally well written although section 4 would gain in clarity by adding some discussions about how these results, which are specific to the Persistence Fisher kernel, compare to more standard ones. Also, I think that the following points need to be clarified: - Up to equation (4) it seems like the fisher distance involves computing the integral of the square root of product of mixture of gaussians densities. However, starting from paragraph on Computation (line 148), this inner product becomes finite dimensional between vectors in the unit sphere of dimension m-1. This transition seems non-trivial to me, Does it comes from the Fast Gauss Approximation? If that is the case, it should be made more explicit that the remaining theoretical results, from proposition 1 to proposition 4 , are specific to this approximation? - In proposition 1, shouldn't the eigen-decomposition depend on the probability \mu? It seems intuitive that the eigenvalues of the integral operator would depend on \mu which is the distribution of the data points as suggested later on in proposition 3 and 4. However, the result is derived for the uniform probability over the sphere, and later on in propositions 3 and 4, the same expressions for the eigenvalues are used for more generic \mu. Propositions 3 and 4 deserve more explanation. How this result compares to standard one in the literature? In equation (9) the Radamacher complexity seems to be defined for the set of unit ball RKHS functions with an additional constraint on the second moment of f under \mu, what justifies such choice of set?

Reviewer 3



A key issue in using persistence diagrams as a topological summary in machine learning applications is define a positive definite inner product structure or kernel based on topological summaries. In particular using the persistence diagram is a problem as straightforward specifications of kernels based on persistence diagrams are not positive definite. In this papers the authors provide an approach that defines a positive definite kernel for persistence diagrams. This is useful and a genuine contribution. There are a few concerns I have mostly in exposition. 1) Typically one starts an information geometric formulation with a likelihood and then the Fisher information matrix is defined based on this likelihood, this is the statistical perspective. Also one can think of the manifold of measures or densities parameterized on a Riemannian Manifold. The smoothed diagrams are not a likelihood in the sense of a data generating process. They may be a likelihood for the summary but again this is not stated. The comparison to the Lafferty and Lebanon work is interesting here but again the multinomial/Dirichlet likelihoods are data generating models. 2) The problem with the persistence diagram is fundamentally the space of diagrams with the Wasserstein metric or the bottleneck distance is problematic. It may be good to reference a paper that states the fact that this space does not for example have unique geodesics, they are CAT spaces bounded below by zero. 3) Does this new kernel matter, is it the case that empirically a more naive kernel based on persistence diagrams at the end of the day works just as well and in practice the empirical kernel matrices are positive (semi) definite.